# Applying the Technology Acceptance Model to Elucidate K-12 Teachers' Use of Digital Learning Platforms in Thailand during the COVID-19 Pandemic

**Noawanit Songkram [1,2] and Hathaiphat Osuwan [1,*]**

1   Department of Educational Technology and Communication, Faculty of Education, Chulalongkorn University, Bangkok 10330, Thailand; noawanit.s@chula.ac.th
2   Learning Innovation for Thai Society Research Unit (LIFTS), Chulalongkorn University, Bangkok 10330, Thailand
*   Correspondence: o.hathaiphat@gmail.com

**Abstract:** With the emergence of the novel coronavirus disease 2019 (COVID-19) pandemic in 2020, educational institutions had to rapidly adapt from face-to-face to online learning to ensure continued education. Various digital learning platforms were tools for online teaching. Therefore, this study aimed to examine the acceptance of technology and behavioral intentions to use digital learning platforms of K-12. Toward this end, the study employed the technology acceptance model as a framework and was expanded with various variables, including technology self-efficacy, subjective norms, and facilitating conditions. An online questionnaire collected data from 519 K-12 teachers in Thailand. The structural equation modeling approach tested the proposed model. The results demonstrated that attitude and subjective norms significantly influenced behavioral intentions toward use, whereas perceived usefulness and ease of use directly influenced attitudes. Moreover, technology self-efficacy, subjective norms, and facilitating conditions influenced perceived ease of use. The findings can serve as a reference for teachers, school administrators, and policymakers in increasing the acceptance of digital learning platforms among future teachers.

**Keywords:** digital learning platform; technology acceptance model; Thai K-12 teachers; attitudes toward technology; structural equation model





## 1. Introduction

Before the emergence of the novel coronavirus disease 2019 (COVID-19) pandemic in 2020, education commonly occurred in physical classrooms. As information and communication technology (ICT) has advanced in recent years, certain technologies, such as blogs, wikis, podcasts, and social bookmarking, have been used in education [1]. However, not all teachers use technology in teaching, such that only tech-savvy teachers are capable of using them [2,3]. Miranda and Russell [2] stated that teachers who have more experiences with technology view technology as a valuable instructional tool and tend to use it in classrooms.

COVID-19 has raised the bar on education. Virtual classrooms have taken over after school closures to prevent the spread of the virus [4]. For example, An et al. [5] examined K-12 teachers in the United States and found that they used various technologies, including online learning platforms and tools (e.g., Seesaw, Schoology, and Quizlet) and online communication and meetings (e.g., Google Meet, MS Teams, and Twitter). In a related study, Slovakian teachers used different pedagogies during remote teaching. In particular, they primarily used EduPage to provide materials and communicate with students [6]. In Cyprus, secondary school teachers used different educational tools (e.g., WhatsApp, Facebook, and Google Classroom) in distance learning [7], whereas elementary school teachers in China used various online teaching platforms, including social tools (e.g., QQ and WeChat), communication tools (e.g., Zoom and Xiaoyu), and platform services (e.g., Seewo

and Xueleyun) [8]. However, most teachers still lacked knowledge and skills in using technology, which rendered teaching and learning inefficient. Plenty of online learning technologies were also available. Occasionally, they create difficulties, such as issues with installation, login problems, download errors, and problems with audio and video.

In addition, personal attention is a significant issue in online learning. Students prefer two-way interaction, which can be difficult to implement at times. Online learning often pertains to the theoretical content, presenting students with difficulty practicing and learning effectively [9]. However, in this 'new normal' world scenario, online learning is necessary to continue learning activities. A few examples of new technologies used in online education include learning management systems (LMSs; e.g., Google Classroom and Moodle), interactive presentation tools (e.g., Mentimeter and Canva), and video conferencing tools (e.g., Google Meet and Zoom) [10,11]. Thailand also faces these challenges in online education. Consequently, Thailand needs to focus on using technology in education. The next step for online education technology will be a digital learning platform (DLP) [12], a combination of an interactive digital environment and administrative tools that target students and teachers. The objective of a DLP is to provide specific digital learning and teaching materials and content [13]. An essential quality of a platform is its adaptability to the needs of students, which leads to improved and faster learning [14]. A DLP is designed to be open and free to all individuals. Therefore, this recent technology is a new challenge for many countries that need to develop and implement DLPs for effective online teaching and learning.

Thailand has been confronted with the COVID-19 pandemic since mid-March 2020 [15]. In terms of education, K-12 schools in Thailand typically begin a new academic year in mid-May. However, the opening for physical classes was delayed until 1 July 2020, to prevent the spread of COVID-19 [16]. At that time, education was conducted online for at least two months to ensure students continued learning during school closures. Thus, teachers had to learn to manage online learning using various technologies, such as Moodle, Zoom, and Google Classroom.

In 2017, the Thai government issued a national reform plan. The education reform section stated that "transform[ing] education through digital transformation has to develop a 'National Digital Learning Platform' by leveraging digital technology, bringing knowledge and learning methods to schools, students and teachers across the country, especially in remote areas." [17] (p. 223). To achieve this reform, the Ministry of Education (MOE) launched a prototype of a national DLP in 2020 called the Digital Education Excellence Platform (DEEP). Specifically, the DEEP includes online learning tools, such as video meeting links, instructional videos, and assignment links [18]. However, the DEEP does not meet the definition of a DLP, as the MOE claims. According to users, the DEEP is only a central platform for teachers and students to access Google for Education and Microsoft 365 for Education [19]. Moreover, it still has certain technical problems, which rendered the DEEP unsuccessful in Thailand due to its lack of flexibility and difficulty in use. It remains similar to the current products, which did not promote its use among many schools. According to the DEEP website, this platform is only employed by 12.95% of users (see Table 1) [20]. Hence, many schools have created their own DLPs. However, since a DLP is a new concept in Thailand, few teachers feel comfortable using such technology in teaching.

**Table 1.** Number of DEEP users.

| List | Numbers |
| --- | --- |
| Number of DEEP users (teachers and students) | 627,297 |
| Teachers under the Ministry of Education | 549,868 |
| Students in schools under the Ministry of Education | 4,293,998 |
| Total number of teachers and students | 4,843,866 |
| Percentage of DEEP users | 12.95% |

*The Need for the Study*

During the COVID-19 pandemic, education was completely conducted through remote learning and digital learning platforms [21]. In the context of developing countries, including Thailand, DLPs were not well adopted in basic education before the pandemic [22], with numerous teachers being unfamiliar with online teaching altogether [23]. Moreover, since a DLP is a new concept in Thailand, there has been little research on how teachers can effectively adopt this platform. Therefore, it is imperative to understand how teachers are willing and ready to adopt a DLP because online teaching should be effective, and students should receive the same quality of education as they do in physical classrooms.

As previously discussed, the technology acceptance model (TAM) will be used to assess teachers' acceptance of DLPs. Davis [24] proposed the TAM to explain the behavioral intention of technology users to utilize technology. According to the TAM, attitudes toward technology use, perceived usefulness, and perceived ease of use are the key determinants of the user acceptance of technology. Several researchers have examined the factors that influence technology acceptance and DLP use [25,26]. Along this line of research, the objective of the current study is to examine the acceptance of technology and behavioral intention to use DLPs of teachers in Thailand. Moreover, the findings can be beneficial to educators, policymakers, and school administrators in the country.

## 2. Literature Review and Hypotheses

### 2.1. Technology Acceptance Model (TAM)

Davis [24] proposed the TAM as an adaptation of the theory of reasoned action (TRA) [27]. Although the TRA is used to explain any human behavior, the TAM is used to explain the intention of technology users to utilize technology [28]. TAM consists of four variables, namely, perceived usefulness (PU), perceived ease of use (PEU), attitude toward use (ATT), and behavioral intention to use (BI) [24].

Specifically, PU is defined as "the degree to which a person believes that using a particular system would enhance his or her job performance," whereas PEU is "the degree to which a person believes that using a particular system would be free of effort" [24] (p. 320). ATT is an individual's positive or negative feelings with regard to performing a behavior [27]. BI denotes the behavioral tendency to keep using technology in the future. Therefore, BI determines the acceptance of technology [29].

According to the TAM, PU and PEU directly influence ATT. Additionally, PU and ATT directly affect BI, whereas PEU directly predicts PU. In this study, if the teachers believe that using a DLP in their teaching is beneficial, their attitudes and intentions scores will be high. When teachers perceive a DLP as easy to use, this view will positively influence PU and ATT directly and BI indirectly through ATT. Previous studies [28,30,31] supported these relationships. Based on the TAM, the study presents the following hypotheses:

**Hypothesis 1 (H1).** *PU has a positive effect on ATT.*

**Hypothesis 2 (H2).** *PU has a positive effect on BI.*

**Hypothesis 3 (H3).** *PEU has a positive effect on PU.*

**Hypothesis 4 (H4).** *PEU has a positive effect on ATT.*

**Hypothesis 5 (H5).** *ATT has a positive effect on BI.*

However, several critical reviews have reported that the TAM cannot explain the technology acceptance of individuals from a broad perspective [32]. Hence, we expanded the model by adding external variables to improve its predictive validity and better elucidate technology acceptance. Based on the literature, individual, environmental, and technological factors may contribute to the under-utilization of technology for teaching. We

selected technology self-efficacy (TSE), subjective norms (SN), and facilitating conditions (FC) as external variables to better understand the technology acceptance of teachers.

*2.2. Technology Self-Efficacy (TSE)*

According to Bandura [33], self-efficacy is a belief in one's ability to act to achieve specific goals. Self-efficacy is concerned with beliefs about one's capability in various situations with previously acquired skills. Consequently, TSE regards belief in one's ability to use technology [34]. Many studies reported that TSE has a significant impact on PEU [26,34–36] and that individuals with high levels of TSE report that technology is easy to use. Therefore, the study puts forward the following hypothesis:

**Hypothesis 6 (H6).** *TSE has a positive effect on PEU.*

*2.3. Subjective Norms (SN)*

SN are "an individual's perception that most people who are important to him or her think he or she should or should not perform a particular behavior" [27] (p. 302). Thus, colleagues, administrators, and students may influence the decision of a teacher to use educational technology in the classroom [34]. In this research, teachers' perceptions toward DLPs are shaped by significant others, including students, colleagues, school administrators, and the parents of students. In particular, the last group believes that digital technologies could provide new knowledge and learning opportunities for their children. Furthermore, it can help children develop learning competencies, language skills, and social skills [37]. Many studies on technology acceptance have used the effect of SN on the behavioral intention to use, specifically PU and PEU [26,31,38]. Accordingly, this study presents the following hypotheses:

**Hypothesis 7 (H7).** *SN has a positive effect on PU.*

**Hypothesis 8 (H8).** *SN has a positive effect on PEU.*

**Hypothesis 9 (H9).** *SN has a positive effect on BI.*

*2.4. Facilitating Conditions (FC)*

FC are the degree to which an individual believes that organizational and technical resources exist to support the use of the system [39] (p. 453). In particular, Ertmer [40] stated that first-order barriers, such as limited resources in technology implementation environments (e.g., access to hardware and software, technical support, and technical training) influenced teachers' efforts to integrate technology into their teaching. Many studies indicated that FC influences the PEU of teachers [31]. Moreover, several studies found that FC has a significant positive influence on the intention to use technology in teaching [34,41]. Teachers who witness the effective use of technology for instruction are more likely to use it in teaching. Therefore, we proposed the following hypotheses:

**Hypothesis 10 (H10).** *FC has a positive effect on PEU.*

**Hypothesis 11 (H11).** *FC has a positive effect on BI.*

**3. Methodology**

*3.1. Data and Sample*

We used a multi-stage sampling design to survey K-12 teachers in Thailand to collect data. The first-stage sampling included purposive sampling to select 12 provinces from four regions in Thailand. The second stage of sample selection selected five schools from each province. The last stage was a random sampling of 10 teachers from each school. The

total number of participants was 519, with a response rate of 86.5%. Before the study, the participants were informed of the research objectives and their right to withdraw from the study at any time. The majority of the participants (31.41%, N = 163) were aged between 26 and 30 years, whereas 52.79% (N = 274), 41.04% (N = 213), and 6.17% (N = 32) taught primary, secondary, and primary and secondary levels, respectively. The greatest number of teachers were from the Thai language department, at 23.89% (N = 124), followed by the Science and Technology Department, at 17.73% (N = 92). The lowest number was from the Occupations Department, at 2.50 % (N = 13).

### 3.2. Instruments

The study employed an online questionnaire presented in two parts. The objective of the first section was to collect personal information, whereas the second section was used to understand the perception of the participants regarding PEU, PU, ATT, BI, TSE, SN, and FC of the DLP. All measurement scales were adopted and adapted from the existing literature [24,31,34,42]. In particular, the items used to measure PU, PEU, ATT, and BI were adapted from Davis [24], whereas the items used to measure TSE were adapted from Wong [42] and Gurer [34]. Moreover, the items used to measure SN and FC were adapted from Teo and Huang [31] and Wong [42]. Table 2 displays the number and sample items for each construct. The questionnaire was composed of 28 items rated using a seven-point Likert-type scale ranging from 1 = strongly disagree to 7 = strongly agree.

**Table 2.** Number of items and sample items for each construct.

| Construct | Items | Sample Items |
| :---: | :---: | :---: |
| PU | 5 | I think using a DLP enhances my online teaching effectiveness. |
| PEU | 4 | A DLP is easy to use. |
| ATT | 4 | I enjoy using DLP. |
| BI | 4 | I am sure I will use a DLP in the future. |
| TSE | 3 | I can solve problems that arise on a DLP by myself. |
| SN | 4 | My colleagues think that I should use a DLP. |
| FC | 4 | When I encounter difficulties in using a DLP, specialized instruction is available to me. |

### 3.3. Data Analysis

We used Cronbach's alpha ($\alpha$) to examine the reliability of each item, which produced scores that exceeded 0.70 for each case. These results indicate that the measures were reliable and internally consistent [43]. We then conducted composite reliability (CR) and average variance extracted (AVE) analyses for convergent validity. Thus, we obtained CR values greater than 0.70 for all measurement questions. In addition, the AVE values, greater than 0.50, indicate that the measurement questions accurately reflected the characteristics of each variable in the model [44]. In addition, discriminant validity was used to measure the extent to which constructs are distinct from one another [43]. Specially, if the square root of the AVE is greater than the correlation coefficients, then the test is deemed valid [44]. Overall, SPSS 22 was used to compute the descriptive statistics, including the mean and standard deviation for all the variables.

The main statistical method for examining the proposed hypotheses was structural equation modeling (SEM), whereas LISREL 8.80 was used to analyze the research model. We found that the indices indicated fit robustness in the SEM analysis. The goodness-of-fit (GFI) indices used in the study included the normed fit index (NFI), comparative fit index (CFI), adjusted goodness-of-fit index (AGFI), root mean square residual (RMR), and root mean square error of approximation (RMSEA). According to the literature on SEM, data fit is excellent when the values for NFI, CFI, GFI, and AGFI are greater than 0.95. For RMR and RMSEA, an excellent data fit requires values of less than 0.05, whereas an acceptable fit requires values of less than 0.08 [43,45].

## 4. Results

To examine the DLP acceptance of K-12 teachers in Thailand, we utilized SEM to illustrate the effect relationship of the variables by analyzing the questionnaire responses. We developed a measurement model for assessing the convergent validity of the constructs and a structural model for examining the direct and indirect impacts of the BI of the participants to use technology.

### 4.1. Descriptive Statistics

Table 3 presents the descriptive statistics for all variables. Overall, the means of all the constructs exceed the midpoint of the scale at 3.50.

**Table 3.** Descriptive statistics of the research constructs.

| Construct | Mean | SD |
|---|---|---|
| PU | 4.652 | 1.112 |
| PEU | 4.514 | 0.812 |
| ATT | 4.486 | 0.951 |
| BI | 4.619 | 1.107 |
| TSE | 4.490 | 1.210 |
| SN | 4.369 | 1.170 |
| FC | 4.473 | 1.206 |

### 4.2. Analysis of Measurement Model

The study assessed the acceptability of the measurement model using the reliability of individual items, the internal consistency between the items, the convergent validity, and the discriminant validity of the model. Table 4 presents the factor loading values, Cronbach's alpha, CR, and AVE of the constructs. Cronbach's alpha in all the variables was found to be greater than 0.70. Therefore, it is satisfactory. Additionally, the CR values reached the minimum threshold of 0.70, ranging from 0.734 to 0.901, suggesting that all the structures can be considered. Meanwhile, the AVE values for all the constructs were between 0.597 and 0.892. The results exceeded a minimum threshold of 0.50, which is acceptable for convergent validity.

**Table 4.** Construct reliability and validity of the measurement model.

| Construct | Items | Loadings | $\alpha$ | CR | AVE |
|---|---|---|---|---|---|
| PU | PU1 | 0.859 | 0.930 | 0.901 | 0.645 |
| | PU2 | 0.850 | | | |
| | PU3 | 0.780 | | | |
| | PU4 | 0.782 | | | |
| | PU5 | 0.739 | | | |
| PEU | PEU6 | 0.883 | 0.835 | 0.851 | 0.597 |
| | PEU7 | 0.922 | | | |
| | PEU8 | 0.604 | | | |
| | PEU9 | 0.627 | | | |
| ATT | ATT10 | 0.737 | 0.816 | 0.860 | 0.613 |
| | ATT11 | 0.576 | | | |
| | ATT12 | 0.946 | | | |
| | ATT13 | 0.827 | | | |
| BI | BI14 | 0.775 | 0.912 | 0.901 | 0.696 |
| | BI15 | 0.915 | | | |
| | BI16 | 0.828 | | | |
| | BI17 | 0.812 | | | |
| TSE | TSE18 | 0.765 | 0.904 | 0.734 | 0.892 |
| | TSE19 | 0.945 | | | |
| | TSE20 | 0.851 | | | |
| SN | SN21 | 0.899 | 0.929 | 0.889 | 0.670 |
| | SN22 | 0.915 | | | |
| | SN23 | 0.753 | | | |
| | SN24 | 0.684 | | | |
| FC | FC25 | 0.713 | 0.914 | 0.864 | 0.616 |
| | FC26 | 0.730 | | | |
| | FC27 | 0.838 | | | |
| | FC28 | 0.848 | | | |

Table 5 demonstrates that all the square roots of the AVE values (bold numbers in the diagonal) are greater than the squared correlations of the paired constructs, indicating that the measurements have adequate discriminant validity.

**Table 5.** Discriminant validity of the measurement model.

|      | PU    | PEU   | ATT   | BI    | TSE   | SN    | FC    |
|------|-------|-------|-------|-------|-------|-------|-------|
| PU   | **0.803** |       |       |       |       |       |       |
| PEU  | 0.661 | **0.773** |       |       |       |       |       |
| ATT  | 0.692 | 0.621 | **0.783** |       |       |       |       |
| BI   | 0.689 | 0.604 | 0.739 | **0.818** |       |       |       |
| TSE  | 0.559 | 0.558 | 0.546 | 0.665 | **0.944** |       |       |
| SN   | 0.596 | 0.482 | 0.583 | 0.731 | 0.647 | **0.819** |       |
| FC   | 0.487 | 0.392 | 0.386 | 0.512 | 0.499 | 0.520 | **0.785** |

### 4.3. Analysis of the Structural Model

Before interpreting the paths of the structural model, we inspected the model fit. The absolute fit indexes were used to determine how well the model fits the empirical data. As indicated in Table 6, the goodness-of-fit indexes provided sufficient levels of fit for the model.

**Table 6.** Fit indexes for the proposed model.

| Fit Indexes | Level of Acceptance Fit | Model | Results |
|-------------|-------------------------|-------|---------|
| $\chi^2$—test | Non-significant | $\chi^2 = 197.707$, df = 253, $p = 0.996$ | Pass |
| $\chi^2/\text{df}$ | <2.00 | 0.781 | Pass |
| CFI | ≥0.95 | 1.000 | Pass |
| GFI | ≥0.95 | 0.969 | Pass |
| AGFI | ≥0.95 | 0.950 | Pass |
| NFI | ≥0.95 | 0.994 | Pass |
| SRMR | <0.05 (good fit) <0.08 (fair fit) | 0.011 | Pass |
| RMSEA | <0.05 | 0.000 | Pass |

### 4.4. Testing the Hypotheses

After validating the measurement model, SEM was used to test the hypothesized model using the entire set of samples (Table 7; Figure 1). Nine out of the eleven hypotheses were confirmed. In particular, the results demonstrated that PEU ($\beta = 0.660$, $p < 0.001$) and SN ($\beta = 0.30$, $p < 0.001$) significantly influence PU. Therefore, H3 and H7 are supported. Moreover, TSE ($\beta = 0.409$, $p < 0.001$), SN ($\beta = 0.286$, $p < 0.001$), and FC ($\beta = 0.170$, $p < 0.01$) significantly influenced PEU. Hence, H6, H8, and H10 are supported. PU ($\beta = 0.590$, $p < 0.001$) and PEU ($\beta = 0.355$, $p < 0.001$) significantly influenced ATT. Therefore, H1 and H4 are supported. ATT ($\beta = 0.834$, $p < 0.001$) and SN ($\beta = 0.476$, $p < 0.001$) significantly influenced BI. Therefore, H5 and H9 are supported. However, since BI exerted non-significant effects on PU and FC, H2 and H11 are rejected.

**Table 7.** Results of hypothesis testing.

| Hypotheses | Relationships | Direction | Path Coefficient | t-Value | Results |
|---|---|---|---|---|---|
| H1 | PU → ATT | Positive | 0.590 | 5.216 *** | Supported |
| H2 | PU → BI | Positive | −0.277 | −1.754 | Not supported |
| H3 | PEU → PU | Positive | 0.660 | 9.544 *** | Supported |
| H4 | PEU → ATT | Positive | 0.355 | 3.183 ** | Supported |
| H5 | ATT → BI | Positive | 0.834 | 5.422 *** | Supported |
| H6 | TSE → PEU | Positive | 0.409 | 5.368 *** | Supported |
| H7 | SN → PU | Positive | 0.30 | 4.956 *** | Supported |
| H8 | SN → PEU | Positive | 0.286 | 3.544 *** | Supported |
| H9 | SN → BI | Positive | 0.476 | 6.942 *** | Supported |
| H10 | FC →PEU | Positive | 0.170 | 2.833 ** | Supported |
| H11 | FC → BI | Positive | 0.010 | 0.217 | Not supported |

** $p < 0.01$. *** $p < 0.001$.

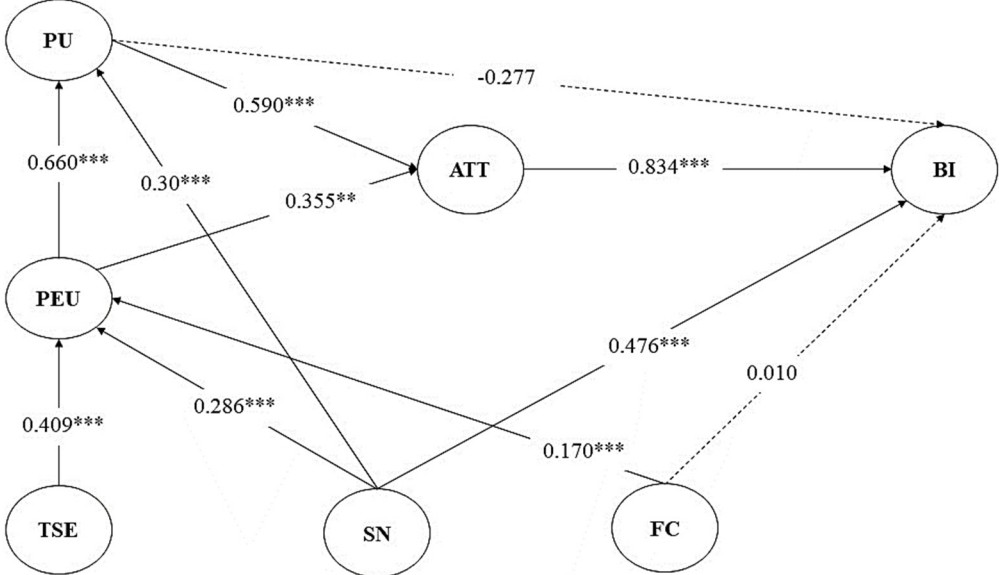

**Figure 1.** Structural model of the DLP acceptance. ** $p < 0.01$. *** $p < 0.001$.

## 5. Discussion

This study aimed to examine the DLP acceptance of K-12 teachers in Thailand and found that nine out of eleven potential relationships were statistically significant ($p < 0.01$, $p < 0.001$). This study demonstrates three significant points discussed below.

First, ATT is a crucial predictor of BI. Specifically, the findings demonstrated that ATT directly predicts BI, which is consistent with the results of previous research [22,46]. Moreover, PU and PEU were predictors of ATT [31,41]. In other words, when teachers perceive that a DLP is useful and easy to use in teaching, they are likely to hold positive attitudes toward DLP use. Moreover, PU and FC had indirect impacts on BI through ATT. These results suggest that although teachers perceive the usefulness of a DLP, they may not necessarily want to use a DLP. Due to the COVID-19 pandemic, most teachers in Thailand adopted DLPs for the first time after online learning replaced face-to-face teaching and learning. This result can be explained through teachers' attitudes toward DLP use. Tondeur et al. [47] found that attitudes toward ICT in education can have a significant influence on educational ICT use. Even if teachers consider DLP useful and schools provide sufficient facilitation, teachers may refrain from DLP use if they lack a favorable attitude toward it. Teachers who are more open to instructional technology are more likely to implement new technology and participate in professional learning activities [48]. Moreover, Drossel

et al. [49] found that positive views of ICT use positively impact the frequency of technology use. Therefore, teachers' attitudes are critical to the acceptance of DLP use.

Second, SN is an important predictor of the acceptance of DLP use because the study found that SN directly influences BI, PU, and PEU. These results are consistent with Mazman Akar [50] and Milutinovic [51], who illustrated that SN significantly influences PU, PEU, and BI. Moreover, this finding indicated that when teachers display high levels of SN, they realize that DLP is useful for teaching and learning. Gurer [34] stated that social pressure and ambition to obtain power and status among colleagues can shape their perceptions of the usefulness and ease of use of technology in a classroom. In general, teachers agree with students or school administrators on whether they consider technology useful and believe that it should be employed in teaching [31]. Moreover, this finding suggested that those who are significant to teachers, including school administrators, colleagues, students, and parents of students, influenced teachers' perception about the ease of using technology in the classroom [52]. Therefore, the results implied that social influence could shape how individuals embrace and use technology [53].

Third, the results illustrated that all the external variables (i.e., TSE, SN, and FC) influenced PEU. Moreover, TSE was found to be the most influential factor on PEU ($\beta = 0.409$; $p < 0.001$). This result is consistent with prior studies [34,54]. Al-awidi and Alghazo [55] found that TSE plays a significant role in elementary school teachers' adoption of technology in the classroom. This finding indicated that teachers may have believed that they possessed different abilities to operate DLPs, such as making announcements, adding videos, and creating tests. This high level of TSE may have positively contributed to the perceptions of the teachers regarding the ease of use of DLPs. SN also significantly predicts PEU ($\beta = 0.286$; $p < 0.001$) under the findings of Gurer [34] and Milutinovic [51]. Moreover, FC ($\beta = 0.170$; $p < 0.001$) positively influenced PEU, which is in line with the literature [41,56]. Li et al. [57] found that if primary and secondary school teachers' PEU in a facilitating, supportive environment is high, they will most likely have a positive attitude toward technology use. Accordingly, when teachers perceive that adequate technical support, such as good quality of technology and technicians, is available, they are more likely to consider technology use to be effortless and become less concerned with technical issues.

## 6. Conclusions

With the increased significance of DLP use in teaching and learning, the study investigated the acceptance of K-12 teachers in Thailand regarding DLP use. The study enhanced the TAM by including technology self-efficacy, subjective norms, and facilitating conditions to explain the behavioral intentions of the participants to use technology.

The attitudes toward use of DLPs ($\beta = 0.834$) and subjective norms ($\beta = 0.476$) were found to be statistically significant ($p < 0.001$) as predictors of behavioral intentions. One of the significant results is that attitudes are crucial. Although teachers perceive the usefulness and ease of using technology or of obtaining sufficient technical support, they may not utilize DLPs if they hold negative attitudes toward technology use. Moreover, social environments may shape the behavioral intentions of teachers. In other words, people who are important and influential to them may influence how they use technology. Therefore, improving teachers' attitudes toward technology is critical for increasing their willingness to use DLPs in classrooms.

## 7. Implications for Research

This research presents essential implications for educators, school administrators, and policymakers. The results indicate that ATT for DLPs is critical, such that school administrators and policymakers should help teachers adjust their attitudes toward the use of new technology [58]. Additional support, such as educational resources, well-established training, technical support, and encouragement should be provided to illustrate that adopting new technology can be both simple and beneficial [59].

Moreover, the TSE of teachers indirectly influenced their BI. Thus, teachers should develop ICT skills to help gain TSE. This process can be achieved by participating in online community practices, such as blogs to share teaching and learning experiences [60], or taking online learning courses. Furthermore, school administrators can help teachers develop TSE by providing professional development for fundamental and advanced ICT skills [61].

## 8. Limitations and Future Research

This research includes some limitations, which represent opportunities for future research. First, we examined three human-related variables: TSE, SN, and FC. However, we overlooked how system design can influence the BI of teachers toward DLP use. Thus, future studies should integrate system design factors, such as system, information, and service quality, to elucidate teachers' intentions toward DLP use. Second, the current study is quantitative and intended to identify the factors of teachers' technology acceptance. Thus, future researchers should conduct qualitative studies to provide an in-depth understanding of why and how attitudes influence the intention of teachers to use DLPs.

**Author Contributions:** Conceptualization, N.S. and H.O.; methodology, H.O.; software, H.O.; validation, N.S. and H.O.; formal analysis, H.O.; investigation, H.O.; resources, H.O.; data curation, N.S. and H.O.; writing—original draft preparation, H.O.; writing—review and editing, N.S. and H.O.; visualization, H.O.; supervision, N.S.; project administration, H.O.; funding acquisition, N.S. All authors have read and agreed to the published version of the manuscript.

**Funding:** This research was funded by the Thailand Science research and Innovation Fund Chulalongkorn University grant number [CU_FRB65_soc(3)_003_27_03].

**Institutional Review Board Statement:** The study was conducted in accordance to the Declaration of Helsinki, the Belmont report, CIOMS guidelines and the Principle of the International Conference on Harmonization—Good clinical practice (ICH-GCP), and approved by the Research Ethics Review Committee for Research Involving Human Subjects: The Second Allied Academic Group in Social Sciences, Humanities and Fine and Applied Arts at Chulalongkorn University (Research Project No. 650030; Approval Date: 1 March 2022).

**Informed Consent Statement:** Informed consent was obtained from all the subjects involved in the study.

**Data Availability Statement:** Data can be requested upon from the researchers.

**Conflicts of Interest:** The authors declare no conflict of interest.

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
