# Peer review of "Applying the Technology Acceptance Model to Elucidate K-12 Teachers’ Use of Digital Learning Platforms in Thailand during the COVID-19 Pandemic"

_sustainability, doi:10.3390/su14106027_

Round 1

Reviewer 1 Report

Paper has too long title.

There is a need to state focus of the paper more clear. In this respect, the section in the introduction can be added as "the need for the study". 

Author Response

Point 1: Paper has too long title.

Response 1: Title’s changed to “Applying the Technology Acceptance Model to Elucidate K-12 Teachers’ Use of Digital Learning Platforms in Thailand During the COVID-19 Pandemic”

Point 2: There is a need to state focus of the paper more clear. In this respect, the section in the introduction can be added as "the need for the study".

Response 2: Added 1.1 The need for the study in the line: 98-106

“During the COVID-19 pandemic, education was completely conducted through re-mote learning and digital learning platforms [21]. In the context of developing countries, including Thailand, DLPs were not well adopted in basic education before the pandemic [22], with numerous teachers being unfamiliar with online teaching altogether [23]. More-over, since a DLP is a new concept in Thailand, there has been little research on how teachers can effectively adopt this platform. Therefore, it is imperative to understand how teachers are willing and ready to adopt a DLP because online teaching should be effective, and students should receive the same quality of education as they do in physical class-rooms.”

Reviewer 2 Report

In general, the article is of excellent quality, although some improvements are suggested to be introduced:

1.- In the lines 31-32 , You say --> "However, not all teachers use technology in teaching, such that only tech-savvy teachers are capable of using them."

Please, it would be necessary to justify this with citations and references that argue it

2.- It is commented that the questionnaire (28 items) has been taken from the literature [22, 33].
Please, could you specify more? Could you indicate the items you have taken? Could you provide more detail on what the questionnaire used consisted of?

3.- The conclusions need to be clearer and more forceful. Perhaps it would help here to talk about the terms investigated again without naming them with the acronym.

Author Response

Point 1: In the lines 31-32 , You say --> "However, not all teachers use technology in teaching, such that only tech-savvy teachers are capable of using them." Please, it would be necessary to justify this with citations and references that argue it.

Response 1: Added two references in the line 34-38:

“However, not all teachers use technology in teaching, such that only tech-savvy teachers are capable of using them [2,3]. Miranda and Russell [2] stated that teachers who have more experiences with technology view technology as a valuable instructional tool and tend to use it in classrooms.”

Point 2: It is commented that the questionnaire (28 items) has been taken from the literature [22, 33].
Please, could you specify more? Could you indicate the items you have taken? Could you provide more detail on what the questionnaire used consisted of?

Response 2: Added more details in the line 201-204:

In particular, the items used to measure PU, PEU, ATT, and BI were adapted from Davis [24], whereas the items used to measure TSE were adapted from Wong [42] and Gurer [34]. Moreover, the items used to measure SN and FC were adapted from Teo and Huang [31] and Wong [42].

Point 3: The conclusions need to be clearer and more forceful. Perhaps it would help here to talk about the terms investigated again without naming them with the acronym.

Response 3: Added more details in the line 329-344:

“With the increased significance of digital learning platform use in teaching and learning, the study investigated the acceptance of K–12 teachers in Thailand regarding digital learning platform use. The study enhanced the TAM by including technology self-efficacy, subjective norms, and facilitating conditions to explain the BI behavioral intentions of the participants to use technology.

The attitudes toward use for digital learning platform (β = 0.834) and subjective norms (β = 0.476) were found to be statistically significant (p <.001) as predictors of behavioral intentionsBI. One of the significant results is that attitudes is crucial. Although teachers perceive the usefulness and ease of using technology or obtain suffi-cient technical support, they may not utilize digital learning platforms if they hold negative attitudes toward technology use. Moreover, social environments may shape the behavioral intentions of teachers. In other words, people who are important and influ-ential to them may influence how they use technology. Therefore, improving teachers’ attitudes toward technology is critical for increasing their willingness to use digital learning platforms in classrooms.”

Reviewer 3 Report

First of all, thank you very much for allowing me to review your article, which I consider to be of interest to the scientific community.

Regarding the introduction, I think that in recent times online teaching derived from the COVID 19 pandemic has been a highly studied construct by the scientific community, so I think it would be of interest to incorporate a greater number of references about this issue as I think it would greatly enrich the importance and scientific evidence on this issue.

Regarding the methodology, the sample used is a large sample for this type of study and the validity and reliability of the instrument used are correctly described.

I think it would be interesting to make comments and descriptions of the tables in a more concise way and not just showing the tables.

It would also be of interest, as well as the introduction, to carry out a more complete discussion of the findings with other authors, since they would support the results obtained more firmly.

Regarding the conclusions and the implications and limitations, I consider that they have been correctly described.

Author Response

Point 1: Regarding the introduction, I think that in recent times online teaching derived from the COVID 19 pandemic has been a highly studied construct by the scientific community, so I think it would be of interest to incorporate a greater number of references about this issue as I think it would greatly enrich the importance and scientific evidence on this issue.

Response 1: Added references in the line 40-50:

“For example, An et al. [5] examined K-12 teachers in the United States and found that they used various technologies, including online learning platforms and tools (e.g., See-saw, Schoology, and Quizlet) and online communication and meetings (e.g., Google Meet, MS Teams, and Twitter). In a related study, Slovakian teachers used different pedagogies during remote teaching. In particularly, they primarily used EduPage to provide materials and communicate with students [6]. In Cyprus, secondary school teachers used different educational tools (e.g., WhatsApp, Facebook, and Google Classroom) in distant learning [7] whereas, elementary school teachers in China used various online teaching platforms, including social tools (e.g., QQ and WeChat), communication tools (e.g., Zoom and Xiaoyu), and platform services (e.g., Seewo and Xueleyun) [8].”

Point 2: I think it would be interesting to make comments and descriptions of the tables in a more concise way and not just showing the tables.

Response 2: Added descriptions of Table 4 in the line 244-249:

“Cronbach’s alpha in all the variables was found to be greater than 0.70. Therefore, it is satisfactory. Additionally, the CR values reached the minimum threshold of 0.70, ranging from 0.734 to 0.901, suggested that all the structures can be considered. Meanwhile, the AVE values for all the constructs were between 0.597 and 0.892. The results exceeded a minimum threshold of 0.50, which is acceptable for convergent validity.”

Added descriptions of Table 5 in the line 252-254:

“Table 5 demonstrates that all the square roots of the AVE values (bold numbers in the diagonal) are greater than the squared correlations of the paired constructs indicated that the measurements have ade-quate discriminant validity.”

Added descriptions of Table 6 in the line 257-258:

“The absolute fit indexes were used to determine how well the model fits the empirical data.”

Point 3: It would also be of interest, as well as the introduction, to carry out a more complete discussion of the findings with other authors, since they would support the results obtained more firmly.

Response 3: Added more discussion in the line 293-294:

“Tondeur et al. [47] found that attitudes toward ICT in education can have a significant in-fluence on educational ICT use.”

In the line 304-306:

“Gurer [34] stated that social pressure and ambition of obtaining power and status among colleagues can shape their perceptions of the usefulness and ease of use of technology in a classroom.”

In the line 315-317:

“Al-awidi and Alghazo [55] found that TSE plays a significant role in elementary school teachers' adoption of technology in the classroom.”

In the line 323-325:

“Li et al. [57] found that if primary and secondary school teachers’ PEU in a facilitating supportive environment is high, they will most likely have a positive attitude toward technology use.”